# Evolution of the Output–Workforce Relationship in Primary Care Facilities in China from 2009 to 2017

**DOI:** 10.3390/ijerph17093043

**Published:** 2020-04-27

**Authors:** Shan Lu, Liang Zhang, Niek Klazinga, Dionne Kringos

**Affiliations:** 1School of Medicine and Health Management, Tongji Medical College, Huazhong University of Science and Technology, Wuhan 430030, China; shanlu@hust.edu.cn; 2Research Centre for Rural Health Service, Key Research Institute of Humanities & Social Sciences of Hubei Provincial Department of Education, Wuhan 430030, China; 3Amsterdam UMC, department of Public Health, Amsterdam Public Health research institute, University of Amsterdam, 1071 JA Amsterdam, the Netherlands

**Keywords:** primary care, workforce, output, latent class growth analysis, China

## Abstract

This study evaluates trends in workforce supply compared with those in the volume of service delivery (output) for basic clinical care (CC) and public health (PH) services from 2009 to 2017 in China. A cross-sectional survey (2018) was combined with retrospective data (2009–2017) from 785 primary care (PC) facilities in six provinces. Measures for the output of clinical care and of public health services were aggregated into a single (weighted) index for both service profiles. The output–workforce relationship was measured by its ratio. Latent class growth analysis and logistic regression analysis were applied to classify trajectories and determine associations with facility-level, geographic, and economic characteristics. From 2009 to 2017, the proportion of PC to overall healthcare workforce decreased from 24.25% to 18.57%; the proportion of PH to PC providers at PC facilities increased from 23.6% to 29.5%, while the proportion of PH output increased from 44.3% to 65.9%. Four trajectories of the output–workforce relationship were identified for CC, and five trajectories for PH services of which 85.3% of the facilities showed initially increasing and then slightly decreasing trends. Geographic characteristics impacted different trajectories. The PC workforce falls behind hospital workforce. The expansion in workload of PH services is unbalanced with that of workforce.

## 1. Introduction

Strong primary care requires a comprehensive scope of services, including promotive, preventive, curative and rehabilitative services. This requirement contributes to increased access to healthcare, decreased utilisation of services provided by hospitals and emergency departments and improved health outcomes and equity [1,2,3]. Primary care is therefore a key strategy for strengthening health systems and service delivery in many low-and middle-income countries, such as China [4,5]. The development of primary care services show three stages since the new China was established in 1949. During the first 30 years since then, China’s primary care system contributed to increases in average life expectancy and decreases in communicable, maternal and neonatal diseases [6,7]. Nevertheless, since the radical economic reforms in the 1980s, the government minimized its investment in healthcare systems and treated healthcare as a free-market consumption activity rather than public goods [8]. Market-oriented healthcare system was carried out at the expense of primary care and strengthened the role of hospitals, which reduced accessibility to health services largely due to high financial constraints [6,7,8,9]. In addition, the outbreak of severe acute respiratory syndrome in 2003 revealed the weakness of the public health system.

To address China’s long-lasting problem in healthcare accessibility and new challenges faced by ageing populations, urbanisation and industrialisation, a new healthcare reform was launched in 2009 [10] in which the government prioritised the reinforcement of primary care systems. Two out of five major reform programmes were directly related to primary care, namely, (1) optimising the healthcare service delivery system at grass-roots level and (2) making basic public health services equally available for all. The first involved activities to construct primary care facilities and strengthen workforce supply and competencies. The second re-established the importance of public health services by issuing an essential package that included nine types of basic services, which primary care facilities should freely provide to all residents. In this reform, the government pointed out that primary care facilities should deliver basic clinical care and public health services.

China’s primary care system benefitted from the 2009 healthcare reform. The government increased subsidies for primary care facilities from US$2.8 billion in 2008 to US$26.4 billion in 2017 [11,12]. Constructing primary care facilities achieved the goal of establishing one community health centre per street in urban areas, one township health centre per town and one clinic per administrative village in rural areas [6]. The primary care system in China predominantly consists of community health centres (stations), township health centres and village clinics. The system provides basic clinical care and public health services to a fifth of the world’s population. In contrast to most of the primary care facilities in other countries, community and township health centres in China are equipped with beds, which are mainly used for rehabilitation and inpatient services, respectively. In 2017, the number of primary care workers reached 3.8 million, an increase of 21.4% since 2009 [12,13]. Moreover, improved accessibility to primary care workforce was confirmed [14]. For public health services, the government budget for basic services per individual increased from ¥15 (exchange rate in 2009: RMB¥6.83 to US$ 1.00) in 2009 to ¥50 (exchange rate in 2017: RMB¥6.47 to US$ 1.00) in 2017.

National-level statistics report an increase in total volume of provided clinical care and public health services [6,15,16]. However, China’s primary care system remains underused, and is suffering from a maldistribution of human resources and shortage of primary care providers [7,9,15]. This situation is worrisome given the expansion in delivery workload of public health services in primary care facilities. Knowledge regarding changes in the relationship between output and workforce at the primary care facility level and variation among facilities is lacking. Such knowledge is relevant because it may guide policy strategies to further increase the availability of a comprehensive scope of clinical care and public health services, equity in access and efficiency of primary care facilities. Therefore, this study focuses on changes in the output–workforce relationship (i.e., output to workforce ratio) to verify whether trends in the workforce supply synchronised with those in service output from 2009 to 2017, considering the regional socioeconomic differences in China.

We address the following research questions:How has the profile of basic clinical care and public health services developed?How has the workforce of basic clinical care and public health services at primary care facilities developed in relation to the overall healthcare workforce?How has the output of basic clinical care and public health services at primary care facilities developed?How has the relationship between output and workforce at primary care facilities developed in terms of trajectories?What are the economic, geographical and facility-level characteristics associated with the changes in the relationship between service output and workforce at primary care facilities?

## 2. Materials and Methods 

### 2.1. Study Design and Sample

This study combined a 2018 cross-sectional survey among primary care facilities with a retrospective data collection from the Health Statistical Annual Reports (2009–2017) for each participating primary care facility. The survey questionnaire was complemented and validated by five senior health services researchers and two managers from primary care facilities in China. A multistage stratified cluster sampling was used to determine the survey samples from primary care facilities. First, two sample provinces were selected from eastern, central and western China. Figure 1 displays a map of China and labels the six pilot provinces used in this study. The eastern, central and western provinces in mainland China are depicted in light grey, dark grey and black, respectively. Then, two cities in each province were selected according to gross regional product (GRP) per capita (high and low). Next, given that the number of counties (representing rural areas) was twice that of districts (representing urban areas) in China, two sample counties (with a relatively high and with low GRP per capita) and one sample district were randomly selected from each sample city. In summary, we identified 24 counties and 12 districts. Finally, all township and community health centres and stations in the selected sample counties and districts were included for investigation. 

Table 1 shows the study sample of the primary care facilities. The number of the primary care facilities in the areas during the study period determined the sample size. In total, 785 primary care facilities were surveyed, and their Health Statistical Annual Reports were collected.

### 2.2. Basic Clinical Care Services and Measurement of Workforce Supply and Output

Basic clinical care provided in primary care facilities includes outpatient and inpatient care. Primary care workforce includes clinical care and public health service providers, which consist of physicians, nurses, pharmacists, clinical laboratory technicians and imaging technicians. In this study, we considered the number of full-time-equivalent clinical care providers as the workforce of basic clinical care, which was calculated by the number of primary care workforce minus that of public health service providers in each facility. The longitudinal data for the number of primary care workforce were derived from the Health Statistical Annual Reports from 2009 to 2017, which was downloaded from the National Health Statistics Reporting System. Through this system, each primary care facility is required to submit a yearly report to record its resources and volume of services at the end of each year. Workforce data included permanent staff and full-time staff with temporary contracts, who are assumed to have similar working hours. The workforce data recorded in the yearly report only included the total number of primary care workforce with distinguished job categories (e.g., physicians, nurses and pharmacists), but not clinical care and public health staff. Therefore, data for the number of public health service providers were obtained from the survey, where each primary care facility was requested to report the number of full-time staff who provided public health services per year during 2009–2017. In order to control recall bias, we requested each facility to report the number of staffs supported by written documents, and if they did not have such written evidence, the numbers should be reported by experienced public health services staffs who had worked in the facility surveyed before 2009.

Output is defined as the quantity of primary care services [17]. Rather than deflated value, physical quantity was used to measure the output of basic clinical care services because such information was accurate and complete [17]. Physical quantity measures for outpatient and inpatient care were outpatient visits and total bed-days occupied by discharged patients per year, respectively. Outpatient visits and bed-day data for each year were collected from each facility’s Health Statistical Annual Report. Measures for the output of outpatient and inpatient care were aggregated into a single index of clinical care output using the service standards as weight. Service standards were defined based on the cost of different services. According to an analysis of hospital costs published by the World Health Organization (WHO) [18], the service standard of inpatient care per bed-day equals three times that of outpatient care per visit. Hence, if the standard for an outpatient visit was one, then the service standard for an inpatient bed-day was three.

We employed the output to workforce ratio to measure the output–workforce relationship for clinical care at the facility level from 2009 to 2017.

### 2.3. Public Health Services and Measurement of Workforce Supply and Output

Public health services were provided in primary care facilities according to the defined package of basic public health services. Table 2 shows the categorised packages into six subgroups. Public health workforce in primary care facilities mainly included full-time physicians and nurses, including permanent staff and full-time staff with temporary contracts who are assumed to have similar working hours. The workforce of public health services was measured by the number of public health providers, which was collected from the survey.

To answer research question 2, data for hospital workforce was compared with that of primary care workforce. Hospital workforce included physicians, nurses, pharmacists, clinical laboratory technicians and imaging technicians. The longitudinal data for hospital workforce were derived from the Health Statistical Annual Reports for 2009–2017. In addition, hospitals did not provide basic public health services; thus, clinical care and public health service workforces were not differentiated in the comparison between hospitals and primary care facilities.

Similar to the measurement for clinical care output, physical quantity was used to measure the output of public healthcare. Table 2 presents the quantity measures for each public health service. Accordingly, these measures were aggregated into a single index of public healthcare output that also used service standards as weight. Service standards were defined on the basis of resource required, risk and complexity to provide each service for primary care facilities. The standards intended to support better resource (i.e. human, financial and material) investment and allocation among primary care facilities were published in a government document from Jingan District, Shanghai in 2016 [19]. Over the last ten years, the basic public health services expanded; the latest 14-type package was published in 2017 and implemented in 2018. Therefore, in this study, we treated the 2015 package as the latest. Table 2 presents the standards for each service in the package. Standards for certain services that lacked specific definition in the government document from Jingan District were established through expert consultation with three senior researchers on health services and two managers from primary care facilities. Data for yearly quantity measures for public health services were collected from the Health Statistical Annual Report.

We employed the output to workforce ratio to measure the output–workforce relationship for public health services at the facility level from 2009 to 2017.

### 2.4. Economic, Geographical and Facility-level Characteristics

Data for the economic and geographical characteristics of each primary care facility were obtained from the cross-sectional survey. GRP per capita (in ten thousand yuan) of a county or district was used to measure economic development. Geographical characteristics were determined by region (provinces where primary care facilities are located), rural or urban, landform (plain, plain and hilly, hilly and mountainous and mountainous) and service radius (distance in kilometre from the primary care facility to the farthest family within the catchment area). Facility characteristics were identified from the yearly Health Statistical Annual Report of each primary care facility. Human resources (number of providers), beds (number of beds), infrastructure (service building area in 100 m^2^) and equipment (number of equipment over 10,000 yuan) of each facility were likewise collected. All facility characteristics were standardised by their standard deviation.

### 2.5. Statistical Analysis

Descriptive figures on changes in clinical care and public health service profile, workforce and output were provided. Latent class growth analysis (LCGA) was used to classify the trajectories of the output–workforce relationship into different groups for clinical care and public health services. For these services, the distribution of output to workforce ratio was slightly skewed. To address this aspect, log-transformed data was used as dependent variables for LCGA. We examined the association between groups and various time-stable (i.e. economic and geographical) or time-dependent (i.e. facility-level) characteristics. In terms of the number and shape of trajectories, the best model was determined by Bayesian information criteria (BIC), representation of substantively distinct trajectories of the output–workforce relationship and distribution of cases across classes. LCGA and analysis of time-dependent characteristics were performed using the PROC TRAJ procedure (Jones & Nagin, Pittsburgh, PA, USA), and analyses of time-stable characteristics for clinical care and public health services were carried out using ordered and multinomial logistic regression, respectively. Missing data analysis in the PROC TRAJ procedure were under missing at random assumptions using all available data and robust maximum likelihood estimation. We included primary care facilities with at least two years of data on output–workforce relationships for the PROC TRAJ procedure. We conducted sensitivity analysis among facilities which had complete data for output–workforce relationships from 2009 to 2017. All analyses were conducted in Stata 15.1. Statistical significance was set to two-tailed *p* < 0.05.

### 2.6. Ethics Approval

This study was approved by the ethics committee of Tongji Medical College, Huazhong University of Science and Technology (IORG No: IORG0003571).

## 3. Results

Overall, 785 primary care facilities participated in this study, of which about half were township health centres (*n* = 420), and the others were community health centres/stations (*n* = 365). There was an equal urban–rural divide in practice location. The descriptive characteristics of the primary care facilities included in this study are presented in Appendix A.

### 3.1. Changes in Breadth of Primary Care Service Profile

Policy documents published by the government regarding the function of primary care services [10,20,21] state that basic clinical care refers to the diagnosis and treatment of common or frequently-occurring diseases, including outpatient and inpatient services. From 2009 to 2017, the service profile of basic clinical care provided in primary care facilities was unclear because no specific definition or disease-based list for common or frequently occurring diseases was established. In 2018, this scenario changed when the government published the standardsfor capacity building in delivering services for primary care facilities through a campaign called Delivering Qualified Services at Primary Care Facilities. According to the standards, all primary care facilities should provide emergency services, internal medicine, surgery, community medical services and traditional Chinese medicine services. Several primary care facilities with high capacity could also provide gynecological (obstetric), ophthalmic and otolaryngologic, dental and rehabilitation services. The standards also recommended 66 basic diseases that should be diagnosed and treated in primary care facilities. The breadth of service delivery profile of public health services changed substantially over the years, with types of services increased from six before 2009 to fourteen in 2017. The breadth of public health services profile is shown in detail in Appendix A.

### 3.2. Changes in Primary Care Workforce in Relation to Hospital Staff

Figure 2 shows the changes in healthcare providers from 2009 to 2017. Over the last nine years, the number of providers in hospitals and primary care facilities increased by 103.0% and 44.6%, respectively. The supply of primary care providers gradually increased, whereas the proportion of primary care providers decreased from 24.25% in 2009 to 18.57% in 2017. A similar proportion of primary care workforce was found from the national level data, regardless of the number of workforces from clinics.

Table 3 presents the changes in and proportion of average workforce of clinical care and public health services per primary care facility. Clinical care and public health service providers increased during the 2009–2017 period. On average, a primary care facility had 27.3 clinical care providers in 2017, indicating an increase of 18.7% since 2009. At the same time, 11.4 public health service providers were identified in 2017, indicating an increase of 60.6% since 2009. The proportion of clinical care providers slightly decreased from 76.4% in 2009 to 70.5% in 2017.

### 3.3. Changes in Primary Care Output

Table 4 shows the average output of clinical care and public health services per primary care facility over time. The average outpatient visits and bed-days showed increases of 35.6% and 43.5%, respectively, during the 2009–2017 period. For public health services, the outputs of different services largely changed. Paper and electronic health records showed similar trends and sharply increased over time. The number of vaccinations provided in primary care facilities was relatively stable, whereas the management cases of patients with hypertension, diabetes and severe mental illness continuously increased and then slightly decreased in 2016. The number of healthcare management cases for the vulnerable population (i.e., children, mothers and the elderly) initially increased and then decreased in the recent two or three years. A comparison of outputs of clinical care and public health services shows that the proportion of the latter increased from 44% in 2009 to 66% in 2017.

### 3.4. Trajectories of the Output–Workforce Relationship at Primary Care Facilities

Appendix B provides a descriptive analysis for the trends in the output–workforce relationship for clinical care and public health services. The output to workforce ratios for clinical care was relatively stable, whereas those for public health services sharply increased and then slightly decreased. The number of facilities with missing data for the output–workforce relationship is shown in Appendix A.

#### 3.4.1. Trajectories of the Output–Workforce Relationship for Basic Clinical Care

We examined 2-to 5-class models. BICs indicated that the model fit better with a higher number of classes. However, the 5-class model included a class that only accounts for 1.7% of the samples, which was relatively small to be accepted. Therefore, we selected the 4-class model that identified four trajectories as the best fit on the basis of model fit statistics and proportion of each class. Results presented in this section were similar to that of sensitivity analysis, and we thus decided to present results for the larger sample size. Detailed results for sensitivity analysis can be provided upon request.

Figure 3 shows the trajectories of the output–workforce relationship for clinical care at primary care facilities. The four trajectories remained stable over 2009–2017 albeit at different levels. Among the four classes, trajectory 1 included 11.7% of the sample and reflected the lowest value of output to workforce ratio at primary care facilities. Trajectories 2 to 4 accounted for 39.9%, 31.5% and 16.8% of the samples, respectively.

#### 3.4.2. Trajectories of the Output–Workforce Relationship for Basic Public Health Services

The 5-class model, which identified five trajectories, was selected as the best fitting solution on the basis of BIC (−235.21), representation of substantively distinct trajectories of the output–workforce relationship and distribution of cases across classes. Although BIC favoured the 6-class model (BIC for 6-class was −174.64), increasing the number of classes divided one relatively stable class in the 5-class model (trajectory 5) into two trajectories with similar trends, which did not substantially change the basic nature of the solution. Results presented in this section were similar to that of sensitivity analysis, and we thus decided to present results for the larger sample size. Detailed results for sensitivity analysis can be provided upon request.

Figure 4 illustrates the trajectories of the output–workforce relationship for public health services at primary care facilities. Trajectory 1 (7.7%) was “persistently low” because it retained the lowest value over 2009–2017, despite its slight increase for the first several years. Trajectories 2 (26.1%) and 4 (31.4%) were ‘low-sharply increasing’ and ‘medium-sharply increasing’, respectively. Both trajectories displayed similar trends in the output to workforce ratio with a rapid increase in 2009–2014 and a slight decrease in 2005–2017. Trajectory 3 (14.7%) was ‘high-slightly decreasing’, which reflects that the output–workforce relationship slightly declined from a high to a median level. Trajectory 5 (20.1%) was “persistently high” and implied that the output to workforce ratio was relatively stable, which only slightly increased in the first four years and then slightly decreased.

### 3.5. Economic, Geographical and Facility-level Characteristics Associated with the Changes

Table 5 provides the characteristics associated with the trajectories of the output–workforce relationship for basic clinical care. Table 6 and 7 provide time-stable and time-dependent characteristics associated with the trajectories of the output–workforce relationship for public health services, respectively. For clinical care, western regions (Chongqing and Guizhou, with the highest odd ratios (ORs) of 8.613 and 4.530, respectively) achieved higher output to workforce ratios than other regions. Primary care facilities in urban areas had a higher output–workforce ratio than rural areas. However, areas with larger service radius were less likely to fall under a higher trajectory compared to those with a smaller service radius. The number of facilities with missing data for characteristics variables is shown in Appendix A.

For public health services, primary care facilities in urban areas were less likely to fall under trajectory 2 (low-sharply increasing) compared with trajectory 1 (persistently low). Compared with Shandong, primary care facilities from Guangdong were more likely to fall under trajectory 3. According to time-dependent characteristics, equipment positively related to trajectory 1. Bed positively related to trajectory 2, but negatively related to trajectory 5. In addition, infrastructure and equipment positively related to trajectory 4.

## 4. Discussion

Since the 2009 healthcare reform, the Chinese government has paid more attention to primary care and particularly public health services by expanding the service profile. However, the growth of primary care workforce fell behind that of hospitals during the period of 2009–2017. The output of public health services increased much faster than that of clinical care. Trajectories of individual primary care facilities in terms of the output–workforce relationship for clinical care remained stable, whereas that for public health services displayed an obvious change. Geographical characteristics, such as region and urban–rural, significantly associated with different trajectories.

Given that one important aim of the 2009 healthcare reform was strengthening the primary care system, the development of the primary care workforce contradicted expectations in relation to those of hospitals. The decreasing proportion of primary care providers to overall healthcare providers illustrated a relative strengthening of the hospital sector and a relative weakening of the primary care sector from 2009 to 2017. This finding may be due to the expansion of hospitals and lack of effective incentives to attract and retain primary care providers. The uncontrolled expansion of secondary and tertiary hospitals attracted not only more medical graduates, but also healthcare workers away from primary care facilities [9]. After the drug mark-ups (primary care facilities were allowed a 15% profit margin on drugs) ended, which used to be a critical source of financing for primary care facilities, the government increased its subsidies to cover the deficit. However, the salary level of primary care workers remained much lower than that of hospitals and accounted for only 70% of the national average annual wage for all occupations [22]. In addition, nonfinancial incentives, such as career development opportunities, were weak. Therefore, we recommend considering the development of hospitals in the context of strengthening the overall healthcare system and improving both the financial and nonfinancial incentives for primary care workforce.

Although the output of clinical care increased, the rate of visits to primary care facilities decreased from 62% (2009) to 54% (2017) [22], which is much lower than the value (no less than 80%) recommended by WHO. In other words, additional primary care providers would be necessary if hospitals refer patients to primary care facilities in the future. As for public health services, the breadth of the service profile expanded with the addition of eight types of new services for patients with noncommunicable diseases and the vulnerable population. Results of the present study showed a remarkable increase in the output of these services, consistent with those of previous studies illustrating that the 2009 reform exerted a considerable impact on service coverage and reduced geographical variation of service accessibility [23]. The systematic coverage rates of child and maternal health management nearly reached 90% in 2014 [16], and the annual increase of coverage rate for patients with diabetes was higher than that of the total number of patients with diabetes within the same period [23]. However, the relatively slow development of the public health service workforce (increase of 56.8% since 2009) and rapid growth of output (increase of 232.8% since 2009) indicated that primary care providers experienced increased workloads and that the number of available public health providers remained insufficient [24,25].

A heavy workload is related to poor performance [26]. The rapidly increasing trend in the output–workforce relationship indicates that the quality of public health services may be suffering. The decreasing trends in paper and electronic health record-keeping since 2016 indicate that a ceiling was reached. The government evaluates the performance of public health service delivery using process-oriented indicators (i.e. filling follow-up records and health records) rather than outcome-oriented ones. Therefore, the daily workload of public health service providers may be disproportionally occupied by administrative duties rather than focusing on improving the quality of care and health outcomes. Mortality due to noncommunicable diseases increased from 4.8/1000 in 2008 to 5.7/1000 in 2016, and control rates for specific indicators, such as blood glucose control, remained persistently low [23].

The four stable trajectories of the output–workforce relationship for clinical care illustrate that the increase in clinical care providers is aligned with that of output at primary care facilities. Results show that primary care facilities in western China are more likely to retain a higher clinical output to workforce ratio, which may be due to the shortage in healthcare workforce. Despite the increased number of workforce in this region, health services are also more responsive to the health needs of residents due to the massive improvement in accessibility to healthcare services [14] and health insurance coverage. Therefore, compared with primary care facilities in other regions, the workforce in western China remains inadequate in relation to health needs. Given the difficulty in attracting and retaining healthcare workforce due to its poorer living standard, lower population density and lower geographic accessibility, new technologies such as telemedicine, can be implemented in western China.

This study identifies five distinct groups of primary care facilities according to the trajectories for the output–workforce relationship in public health services. In contrast to those of clinical care, trajectories for public health services show obvious changes in 2009–2017. Groups 2 and 4, which account for over half of the total facilities, present sharply increasing trends in the first six years and then slightly decreasing trends. This finding indicates that the increase in output was initially much faster than that of the workforce, after which the gap gradually diminished. The trends can be attributed to the mandatory requirement of expanding the scope and volume of public health services and strict government supervision for the first several years after the 2009 reform. Groups 3 and 5 with high starting points did not show obvious increases but displayed a slight decrease and a relatively stable trend, respectively. These trends seem to indicate that primary care facilities in China experienced the highest workload for public health services during the past decade, and that the workload for public health providers can gradually become as normal and stable as clinical care in the future. Results show that the trajectories for primary care facilities in urban areas with low starting points are less likely to present sharply increasing trends, indicating a greater likelihood to gain balance between output and workforce compared with facilities in rural areas. The average number of public health workforce per facility in rural areas is less than that in urban areas (i.e., 9.0 and 9.8, respectively) during the 2009–2017 period. Therefore, the workload for public health service providers in rural areas is likely heavier, which may negatively impact the quality of services.

This study analyses clinical care and public health services separately in terms of workforce, output and their relationship at primary care facilities. The average number of public health providers per facility is 11.6 while that of clinical care is 27.9; however, the output of the former is much higher than that of the latter. This finding indicates that public health services are insufficiently valued by primary care facility managers and providers, and that inadequate flexibility is provided for job substitution between clinical and public health providers and between settings. Thus far, the two types of services are relatively fragmented, and providers in charge of different services seldom communicate with one another [23]. For example, public health service providers only conduct disease screening, follow-ups and maintain and update health records for patients with diabetes, whereas clinical care providers provide prescription, medication or referral only when patients approach them to seek care. Given the increasing evidence that better coordination and collaboration between clinical care and public health contributes to improved health [27], the uncooperative work between clinical and public health service providers could be another reason for the adverse health outcomes of patients with chronic diseases, apart from the previously mentioned lack of outcome-driven indicators. Currently, health insurance schemes cover basic clinical care, whereas government subsidies finance basic public health services. Strengthening the synergy between clinical care and public health services is recommended, which may require integrating the fragmented financing systems for the two types of services and encouraging different healthcare providers to work together. Synergy of services is expected to contribute to relieving public health providers of the heavy workload and improving health outcomes. We also recommend introducing an increased managerial flexibility at the primary care facility level to enable care substitution between clinical care and public health providers and settings. Such introduction would likewise render the facilities accountable for the quality of care provided, instead of the volume of care.

This study has its strength and limitations. To our knowledge, this study is the first to investigate and analyse the workforce and output of public health services, and to compare clinical care and public health services in terms of workforce, output and the relationship between them. Most of previous studies on China’s healthcare reform use national or regional level statistics; hence, microcosmic facility-based analysis remains lacking. Therefore, this study set a single primary care facility as the research unit. However, data from one district in Hubei province and one district and one county in Henan province is lacking. Major differences in healthcare policy and human resources typically occur among provinces rather than among districts and counties in China; thus, we assume that the districts or counties with missing data have similar results with those of other sample districts and counties from the same province. In addition, this study treats individual primary care facilities as the research unit and uses average rather than total values for majority of the analyses. Therefore, we assume that missing data have a limited influence on the results. Furthermore, the output of public health consultation activities and health lectures are set as minimum values required by the government for each primary care facility during the 2009–2017 period. Although this aspect might underestimate the output for certain facilities, managers from primary care facilities stated that few facilities exceeded the basic requirement due to the heavy workload for public health services. Finally, the results of the sensitivity analysis for time-dependent characteristics were not consistent with the results showed in Table 7, which may indicate the results were not reliable, and thus we did not interpret the results in our discussion. This would require attention for future research.

The findings of this study might be relevant for other primary care facilities in China as well as low-and middle-income countries with a relatively weak primary care system and similar settings as included in this study. 

## 5. Conclusions

Despite its gradual development, the primary care workforce in China still falls behind hospital workforce due to lack of effective incentives for the former and the expansion of hospitals. Therefore, we recommend considering hospital development in the context of the strengthening of the overall healthcare system and improving both the financial and nonfinancial incentives for primary care providers. The expansion in breadth and volume of the workload of public health services is unbalanced with that of the public health services workforce (which can threaten professionals’ health, and at the same time, quality of care and health outcome) requiring further attention. The provision of clinical care and public health services is relatively fragmented with less flexibility in care substitution between providers and settings. Thus, we recommend policies that focus on reaching a synergy between the two types of services, such as by integrating the fragmented financing systems, increasing managerial flexibility at the facility level to allow for care substitution, and by encouraging models for collaborative care and teamwork. Geographical variations in the evolution of the output–workforce relationship indicates that differentiated policies are needed to strengthen primary care.

## Figures and Tables

**Figure 1 ijerph-17-03043-f001:**
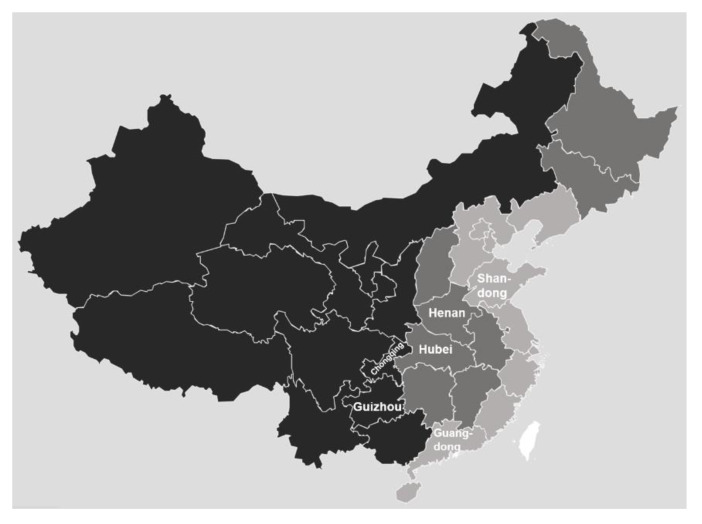
Sample provinces.

**Figure 2 ijerph-17-03043-f002:**
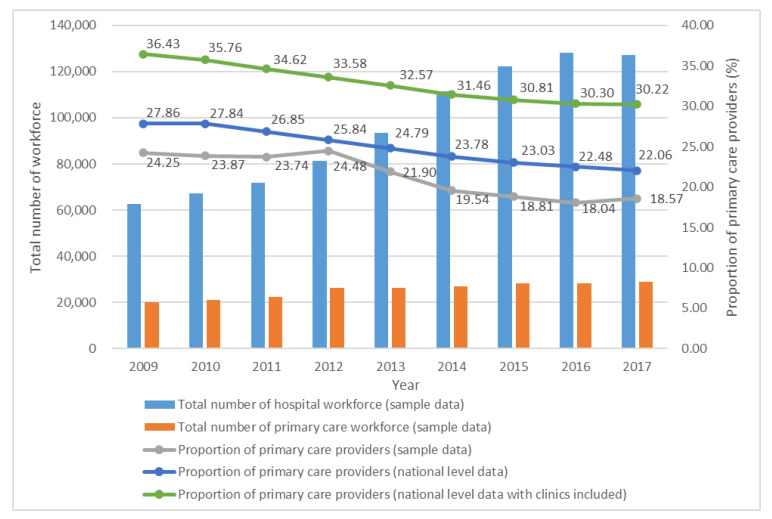
Change in healthcare providers in hospitals and primary care facilities from 2009 to 2017 (*n* = 1100). Note: 1. The primary care workforce in our sample data refers to the workforce from township and community health centres and stations. Data exclude workforce from clinics. 2. The sample includes community health stations that were excluded from the survey because of affiliation to community health centres and lack of independent data. Hence, the sample size is larger than the survey sample.

**Figure 3 ijerph-17-03043-f003:**
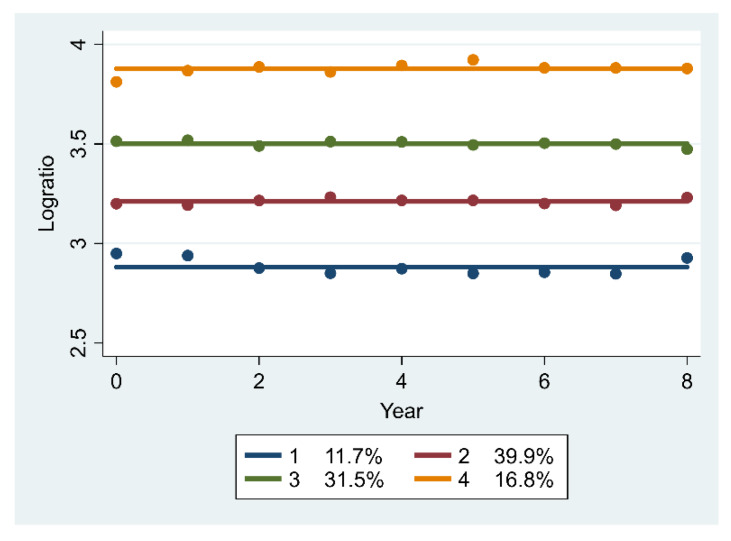
Trajectories of the log-transformed ratio of output to workforce for clinical care (*n* = 589).

**Figure 4 ijerph-17-03043-f004:**
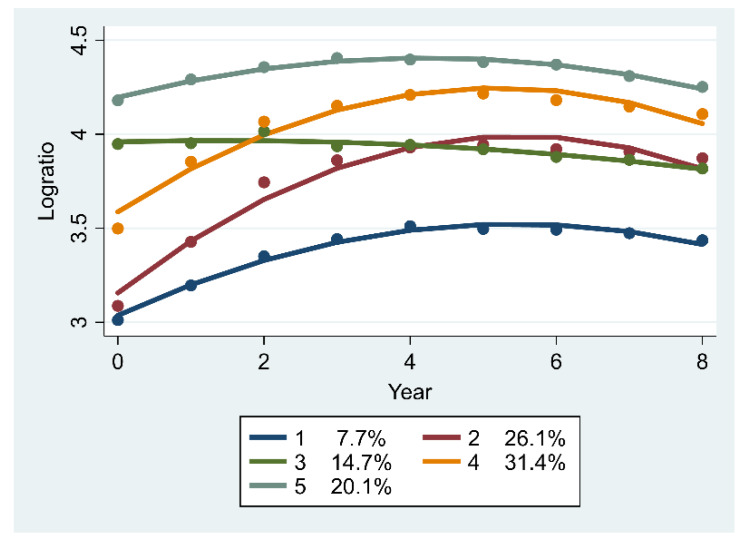
Trajectories of the log-transformed ratio of output to workforce for public health services (*n* = 571).

**Table 1 ijerph-17-03043-t001:** Study sample of primary care facilities.

Region	No. of Provinces	No. of Sample Provinces	Sample Provinces	Sample Cities	Districts and Counties
Eastern China	12	2	Guangdong	Shenzhen	2 districts ^1^
				Shaoguan	4 counties ^1^
			Shandong	Qingdao	1 district; 2 counties
				Jining	1 district; 2 counties
Central China	9	2	Hubei	Yichang	1 district; 2 counties
				Huanggang	1 district ^3^; 2 counties
			Henan	Luoyang	1 district; 2 counties
				Shangqiu	1 district ^3^; 2 counties ^3^
Western China	10	2	Guizhou	Zunyi	1 district; 2 counties
				Tongren	1 district; 2 counties
			Chongqing ^2^	2 districts	2 districts
				4 counties	4 counties

^1^ Shenzhen City does not have a county, and thus we selected two districts from Shenzhen City and four counties from Shaoguan City for Guangdong Province. ^2^ Chongqing is a municipality directly under the Central Government and a provincial-level city; thus, we selected two districts (with relatively high GRP per capita) and four counties (two with relatively high GRP per capita and two with low GRP per capita). ^3^ The Health Statistical Annual Reports (2009–2017) for participating primary care facilities from a district in Huanggang and one district and one county in Shangqiu were missing.

**Table 2 ijerph-17-03043-t002:** Service standards for clinical care and public health services.

Service Categories	Quantity Measures for Output (per Year)	Service Standards	Units for Service Standards
Clinical care			
Outpatient care	Outpatient visits: Total number of outpatient visits within a year for a primary care facility	1 [18,19]	Per visit
Inpatient care	Total bed-days occupied by discharged patients: Total bed-days occupied by discharged patients within a year for a primary care facility	3 [18]	Per bed-day
Public health service			
Health record keeping and updating	Paper health records: Cumulative number of health records of urban and rural residents established in accordance with the government’s requirements	1 ^1^	Per health record
	Standardized electronic health records: Number of electronic health records established in accordance with the government’s requirements	1 ^1^	Per electronic health record
Health promotion	Public health consultation activities: Number of public health consultation activities held by a primary care facility	10 ^1^	Per health consultation activity
	Health lectures: Number of health lectures held by a primary care facility	10 ^1^	Per health lecture
Preventive care	Reported cases of infectious diseases and public health emergencies: Number of cases of infectious diseases and public health emergencies reported by a primary care facility through the Internet, phone or fax	2 [19]	Per case
	Vaccinations: Frequency of vaccinations for children aged 0–6 years according to the National Immunization Program	1 [19]	Per vaccination
Child and maternal healthcare	Number of children healthcare management: Number of children aged 0–6 years who were under health management and provided with related services according to the government’s requirement	8 [19]	Per person per year
	Number of maternal healthcare management: Number of pregnant and parturient woman who were under health management and provided with related services according to the government’s requirement	8 [19]	Per person per year
Disease management	Management cases of patients with hypertension: Number of patients with hypertension who were under health management and provided with related services according to the government’s requirement	8 [19]	Per person per year
	Management cases of patients with diabetes: Number of patients with diabetes who were under health management and provided with related services according to the government’s requirement	8 [19]	Per person per year
	Management cases of patients with severe mental illness: Number of patients with severe mental illness who were under health management and provided with related services according to the government’s requirement	8 [19]	Per person per year
	Management cases of patients with tuberculosis: Number of patients with tuberculosis who were under health management and provided with related services according to the government’s requirement	10 [19]	Per case
Other services	Number of elderly healthcare management: Number of the elderly over 65 years who were under health management and provided with related services according to the government’s requirement	4 [19]	Per person
	Frequency of health supervision and management: Frequency of health supervision, such as patrolling for food-borne diseases, hygiene of drinking water, school hygiene, illegal medical practice and illegal blood collection and supply, etc	5 ^1^	Per time
	Number of traditional Chinese medicine management cases: Number of the elderly over 65 years and children under 6 years who were provided with traditional Chinese medicine health management services, such as constitution identification and healthcare guidance	2 ^1^	Per case

^1^ The authors set the definition for the service standards based on the consultation with health service researchers and managers from primary care facilities.

**Table 3 ijerph-17-03043-t003:** Clinical care and public health service workforce per primary care facility for 2009–2017 (*n* = 785).

Workforce Categories	2009	2010	2011	2012	2013	2014	2015	2016	2017
Workforce_clinical (mean)	23.0	22.8	22.4	25.1	25.7	26.5	27.1	27.4	27.3
Proportion_clinical (%)	76.4	75.0	73.4	74.9	74.3	73.2	72.7	71.7	70.5
Workforce_public (mean)	7.1	7.6	8.1	8.4	8.9	9.7	10.2	10.8	11.4
Proportion_public (%)	23.6	25.0	26.6	25.1	25.7	26.8	27.3	28.3	29.5

**Table 4 ijerph-17-03043-t004:** Average output of clinical care and public health services per primary care facility (*n* = 785).

Services Delivery Output Categories	2009	2010	2011	2012	2013	2014	2015	2016	2017
Clinical care									
Outpatient visits	28,772	30,361	32,250	34,433	37,213	38,516	38,729	38,617	39,004
Total bed-days	4822	4991	5245	6148	6432	6617	7025	6915	6919
Public health services									
Paper health records	12,147	17,476	24,073	24,506	26,321	27,437	30,137	29,095	25,392
Standardized electronic health records	8616	12,334	20,831	22,525	24,795	26,050	28,504	27,740	23,866
Public health consultation activities ^a^	6	6	6	9	9	9	9	9	9
Health lectures ^a^	12	12	12	12	12	12	12	12	12
Reported cases of infectious diseases and public health emergencies	0	0	0	34	30	44	52	44	48
Vaccinations	7171	7212	7587	7726	8359	7416	7968	7166	8247
Number of children healthcare management	0	0	0	1598	2033	2064	2131	2042	2079
Number of maternal healthcare management	203	244	258	320	362	391	537	449	364
Management cases of patients with hypertension	387	698	1184	1468	1584	1602	1766	1646	1762
Management cases of patients with diabetes	92	161	274	379	416	436	517	484	561
Management cases of patients with severe mental illness	12	37	52	64	75	80	164	98	109
Management cases of patients with tuberculosis	1	0	0	0	0	0	17	16	12
Number of the elderly healthcare management	965	1647	1952	2329	2246	2194	2102	2146	1980
Frequency of health supervision and management	0	0	0	30	42	52	82	122	136
Number of traditional Chinese medicine management cases	0	0	0	0	2029	3317	3224	3228	2850
Comparison between clinical care and public health services									
Output of clinical care (mean per primary care facility)	42,485	44,431	46,749	51,906	54,805	56,018	57,452	57,397	58,072
Proportion of clinical care output (%)	56	47	39	37	35	34	32	33	34
Output of public health services (mean per primary care facility)	33,787	49,824	72,367	87,909	103,006	110,038	121,394	117,693	112,438
Proportion of public health service output (%)	44	53	61	63	65	66	68	67	66

a. The outputs of the public health consultation activities and the health lectures were the basic requirements set by the government.

**Table 5 ijerph-17-03043-t005:** Characteristics associated with different trajectories of basic clinical care (*n* = 589).

Characteristics	OR	95% Confidence Interval		*p*-Value
		Lower	Upper	
GRP per capita	0.974	0.940	1.009	0.138
Region: ref-Shandong				
Guangdong	4.438	2.700	7.295	<0.001
Hubei	3.376	1.769	6.443	<0.001
Henan	2.726	1.379	5.390	0.004
Chongqing	8.613	4.846	15.307	<0.001
Guizhou	4.530	2.435	8.426	<0.001
Urban: ref-Rural				
Urban	2.041	1.381	3.018	<0.001
Landform: ref-Plain				
Plain and hilly	1.083	0.667	1.759	0.748
Hilly and mountainous	0.660	0.398	1.093	0.106
Mountainous	0.527	0.302	0.919	0.024
Service radius	0.978	0.963	0.994	0.006

**Table 6 ijerph-17-03043-t006:** Time-stable characteristics associated with different trajectories of basic public health services (*n* = 571).

Characteristics	OR	95% Confidence Interval		*p*-Value
		Lower	Upper	
1. Trajectory 1 (reference)				
2. Trajectory 2				
GRP per capita	1.013	0.883	1.162	0.852
Region: Shandong (reference)				
Guangdong	2.160	0.525	8.882	0.286
Hubei	0.286	0.067	1.215	0.090
Henan	1.359	0.234	7.886	0.733
Chongqing	0.451	0.118	1.717	0.243
Guizhou	1.330	0.297	5.952	0.709
Urban: Rural (reference)				
Urban	0.269	0.099	0.727	*0.010*
Landform: Plain (reference)				
Plain and hilly	1.026	0.289	3.640	0.968
Hilly and mountainous	2.019	0.537	7.596	0.298
Mountainous	1.314	0.343	5.032	0.690
Service radius	1.010	0.969	1.052	0.649
3. Trajectory 3				
GRP per capita	0.950	0.819	1.101	0.493
Region: Shandong (reference)				
Guangdong	5.231	1.205	22.708	*0.027*
Hubei	0.579	0.118	2.834	0.500
Henan	0.318	0.024	4.198	0.384
Chongqing	0.840	0.195	3.612	0.814
Guizhou	0.619	0.108	3.552	0.590
Urban: Rural (reference)				
Urban	1.076	0.349	3.319	0.899
Landform: Plain (reference)				
Plain and hilly	1.057	0.290	3.849	0.933
Hilly and mountainous	1.592	0.380	6.661	0.525
Mountainous	0.874	0.194	3.937	0.861
Service radius	0.990	0.941	1.042	0.704
4. Trajectory 4				
GRP per capita	1.100	0.970	1.248	0.136
Region: Shandong (reference)				
Guangdong	2.116	0.571	7.849	0.262
Hubei	0.445	0.129	1.542	0.202
Henan	1.530	0.295	7.929	0.613
Chongqing	0.584	0.174	1.962	0.384
Guizhou	1.175	0.292	4.731	0.821
Urban: Rural (reference)				
Urban	0.556	0.227	1.363	0.200
Landform: Plain (reference)				
Plain and hilly	1.169	0.389	3.518	0.781
Hilly and mountainous	1.441	0.423	4.913	0.559
Mountainous	0.809	0.233	2.814	0.739
Service radius	1.009	0.970	1.049	0.665
5. Trajectory 5				
GRP per capita	1.075	0.942	1.226	0.285
Region: Shandong (reference)				
Guangdong	1.741	0.428	7.090	0.439
Hubei	0.487	0.123	1.928	0.305
Henan	1.305	0.222	7.679	0.768
Chongqing	0.431	0.111	1.678	0.225
Guizhou	0.351	0.064	1.907	0.225
Urban: Rural (reference)				
Urban	0.489	0.180	1.326	0.160
Landform: Plain (reference)				
Plain and hilly	1.083	0.322	3.640	0.897
Hilly and mountainous	1.458	0.380	5.603	0.583
Mountainous	0.599	0.145	2.477	0.479
Service radius	1.024	0.981	1.068	0.280

**Table 7 ijerph-17-03043-t007:** Time-dependent characteristics associated with different trajectories of basic public health services (*n* = 571).

Characteristics	β	SE	*p*-Value
Time-dependent characteristics			
1. Trajectory 1			
Human resource	0.0002	0.0005	0.773
Beds	0.0042	0.0128	0.744
Infrastructure	−0.0260	0.0229	0.256
Equipment	0.0366	0.0142	0.010
2. Trajectory 2			
Human resource	0.0008	0.0003	0.011
Beds	0.0484	0.0121	0.000
Infrastructure	−0.0010	0.0117	0.934
Equipment	0.0001	0.0091	0.992
3. Trajectory 3			
Human resource	0.0019	0.0008	0.013
Beds	0.0176	0.0094	0.060
Infrastructure	0.0048	0.0094	0.610
Equipment	0.0054	0.0100	0.588
4. Trajectory 4			
Human resource	0.0002	0.0005	0.701
Beds	0.0235	0.0156	0.133
Infrastructure	0.0406	0.0165	0.014
Equipment	0.0439	0.0149	0.003
5. Trajectory 5			
Human resource	0.0002	0.0007	0.731
Beds	−0.0464	0.0179	0.010
Infrastructure	−0.0063	0.0148	0.670
Equipment	−0.0055	0.0107	0.605

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
