# Peer review of "Evolution of the Output–Workforce Relationship in Primary Care Facilities in China from 2009 to 2017"

_ijerph, 2020, doi:10.3390/ijerph17093043_

Round 1
Reviewer 1 Report
This paper presents information about primary care facilities in China. I have some comments and suggestions about this manuscript.
First of all, dear Authors follow the Instruction for Authors one more time.
The abstract should be a single paragraph and should follow the style of structured abstracts, but without headings, and should be a total about 200 words maximum.
Provide reference to this document Health Statistical Annual Reports and add it also in main text.
Second section should be called Materials and Methods.
In section statistical analysis add information about quantity and the qualitative data. Provide all information such as numbers and percentage in section Results not in this section.
Table 3 is incomprehensible and not clear for Readers – in my opinion this table should remove.
In section Results should be only your results of your own research – there is no need to add references in this section, you can use it in discussion.
Table 7 is very big and not clear for readers and you should think about remove of p-value not statistically significant. In table 7 there is OR and analysis of regression – regression should be separate table.
Please analyze your research because your paper is very long, and it have many tables and figures – maybe you could remove some of them and make it shorter.
Authors contribution must be modified due to Instruction for Authors,
The References must be modified due to Instruction for Authors.
Reviewer 2 Report
Dear authors,
As someone who works at public health sector and healthcare analysis, planning and organization, I want to evaluate this work with all flying colors.
I'm pleased to recommend this manuscript for publishing, with eventual proofreading corrections.
Author Response
Dear reviewer,
Thank you very much for recommending our manuscript for publishing and for the willingness to provide eventual proofreading corrections.
Kind regards,
All authors
Reviewer 3 Report
It's an interesting article about the workforce in primary care and the development of primary compared to hospital care. I have just some minor remarks.
Row 162 I don't understand the word 'input' over here. Didn't you measure the output?
Table 3: Types of basic public health services: In my version of the article (*.pdf) the table results weren't visible.
Figure 2: Changes in healthcare etc. I suppose that the number of workforces was expressed as a mean per provider? For example, in 2017 the mean for primary care was 33,5 (table S1)? Or are these figures totals for all providers together?
Table 5: I would suggest to take only whole numbers. The figures after the point give details which aren't relevant.
One of your recommendations is to control the development of hospitals. Is it a realistic strategy? Wouldn't it be wiser to strengthen the financing of primary healthcare?
Table S1: The characteristics aren't always clear: human resource: the mean number per facility? Beds: the mean number? Infrastructure: the mean number of buildings? Rooms? Equipment: Mean of what?
